# Larger Probes Tell a Different Story: Extending Psycholinguistic Datasets Via In-Context Learning

**Namrata Shivagunde, Vladislav Lialin, and Anna Rumshisky**
Department of Computer Science
University of Massachusetts Lowell
{nshivagu,vlialin,arum}@cs.uml.edu

## Abstract

Language model probing is often used to test specific capabilities of models. However, conclusions from such studies may be limited when the probing benchmarks are small and lack statistical power. In this work, we introduce new, larger datasets for negation (NEG-1500-SIMP) and role reversal (ROLE-1500) inspired by psycholinguistic studies. We dramatically extend existing NEG-136 and ROLE-88 benchmarks using GPT3, increasing their size from 18 and 44 sentence pairs to 750 each. We also create another version of extended negation dataset (NEG-1500-SIMP-TEMP), created using template-based generation. It consists of 770 sentence pairs. We evaluate 22 models on the extended datasets, seeing model performance dip 20-57% compared to the original smaller benchmarks. We observe high levels of negation sensitivity in models like BERT and ALBERT demonstrating that previous findings might have been skewed due to smaller test sets. Finally, we observe that while GPT3 has generated all the examples in ROLE-1500 is only able to solve 24.6% of them during probing. The datasets and code are available on Github[1].

## 1 Introduction

Understanding the limitations of large language models (LLMs) becomes ever more important with their accelerated adoption and application to real-life tasks. After the original discovery that large LMs could perform simple NLP tasks without additional training (Radford et al., 2019), the use of these models has rapidly grown, as have their capabilities (Brown et al., 2020; Sanh et al., 2021; Chowdhery et al., 2022). As the community invests considerable effort into creating, training, and deploying these models (Zhang et al., 2022; Black et al., 2021), it is important to understand the types of data and tasks they might not be well-suited.

The field of analysis of pre-trained models has grown rapidly in recent years (Zagoury et al., 2021; Liu et al., 2021; Lialin et al., 2022; bench authors, 2023; Rogers et al., 2020). Methods such as attention pattern analysis (Kovaleva et al., 2019; Kobayashi et al., 2020), linear probing (Tenney et al., 2019), and zero-shot probing (Belinkov et al., 2020; Talmor et al., 2019; Ettinger, 2019; Lialin et al., 2022) allow us to evaluate specific capabilities of pre-trained models. Zero-shot methods give us arguably the most clear picture, as they directly probe what the model learned through the upstream task and allow the researcher to target very specific skills such as understanding of negation or role.

However, even though these methods do not require training data, producing a good dataset for zero-shot evaluation of these language models is not an easy task. We want these datasets to be clean, diverse and to have enough statistical power to be useful for model comparison (Card et al., 2020). Many existing probing datasets struggle with at least one of these requirements.

Psycholinguistic datasets used in a study by Ettinger (2019) have been particularly interesting in that they enabled a comparison between model behavior and human response, including both N400 effects and well-reasoned cloze judgments by human speakers. Despite being used in multiple studies since (Lialin et al., 2022; Rogers et al., 2020; Zhang et al., 2020), these datasets are quite small, ranging in size from 18 sentence pairs in negation (NEG-136-SIMP) to a maximum of 44 sentence pairs in the role-reversal dataset (ROLE-88).

In our work, the NEG-136-SIMP and ROLE-88 datasets are dramatically extended. For each of them, we follow the original dataset collection method (Fischler et al., 1983; Chow et al., 2016), with the exception of N400 amplitude (requires electroencephalography) and cloze probability studies on humans (Federmeier and Kutas, 1999). To explore different approaches to data ex-

---

[1]https://github.com/text-machine-lab/extending_psycholinguistic_dataset

tension, we extended negation dataset using two methods 1) a template-based and 2) using GPT3[2] with human-in-the-loop. For ROLE, we only used GPT3 to extend the dataset, as no additional role categories were available in original source paper for this data (Chow et al., 2016).

To understand how well language models perform on these extended datasets, we evaluated 22 models, including GPT3, following the methodology from Lialin et al. (2022). Compared to the original test sets, we see a significant drop (up to 57%) in accuracy for both Role and Negation tasks. At the same time most models show higher sensitivity to negation compared to the original dataset. Finally, while GPT3 has **generated all of the examples** in ROLE-1500, it is only able to predict the correct answer in **24.6% cases** (top-5 accuracy). ALBERT-v2-xxlarge surpasses GPT3 by 4.6%.

## 2 Related Work

Recent research has shown that LLMs are capable of generating labeled data (Anaby-Tavor et al., 2020; Papanikolaou and Pierleoni, 2020; Kumar et al., 2020; Meng et al., 2022; Mohapatra et al., 2020; Yang et al., 2020; Li et al., 2022; Wu et al., 2022). Most previous studies used GPT or GPT2 for dataset extension (Schick and Schütze, 2021; Gao et al., 2022; Whitfield, 2021; Anaby-Tavor et al., 2020). Smith et al. (2022) attempted to generate labels for the unlabelled dataset using GPT3. In our work, we generate the entire dataset, rather than just the labels. Bonifacio et al. (2022) used GPT3 to generate questions, sampling from 100k documents to create the prompts. Han et al. (2021) and Liu et al. (2022) prompted GPT3 to generate synthetic translation and NLI datasets, respectively.

Lialin et al. (2022) and Ettinger (2019) evaluated language models on smaller datasets for negation and role reversal. We extend these datasets to around 1500 data points and evaluate 22 models, including GPT3. To our knowledge, our work is the first to extend psycholinguistic datasets and use it to evaluate an extended set of models.

## 3 Data generation

### 3.1 Negation Dataset: NEG-1500-SIMP

The existing negation dataset NEG-136-SIMP from Ettinger (2019) consists of 18 sentence pairs. Each pair is made of one affirmative and one negated

sentence e.g. the sentences "A robin is a bird" (affirmative) and "A robin is not a tree" (negated) form a pair. We extend this dataset using two methods: template-based and generation-based. By employing both methods, we could gauge the potential strengths and limitations inherent to each method, which we discuss in Section 5. We create 770 sentence pairs with the template-based method and generate 750 pairs using GPT3. We refer to these datasets as NEG-1500-SIMP-TEMP and NEG-1500-SIMP-GEN, respectively.

**NEG-1500-SIMP-TEMP**   Each pair in the original dataset follows a template. For affirmative sentences, the template is "*A/An {subject} is a/an {object}*". Its negated version is "*A/An {subject} is not a/an {another object}*". Battig and Montague (1969) and Fischler et al. (1983) provide a template and a list of objects (e.g., robin, pine) and the corresponding categories (e.g., bird, tree) which are used to generate samples [3]. For instance, the category, "bird" in the example, "A robin is a bird" is replaced with another category ("tree") to generate the negated example "A robin is not a tree".

We experimented with using WordNet to generate from this template, but found that it required a careful curation of hypernym hierarchies. Since our goal was to reduce human effort and cost, we decided against it. We used a template from Fischler et al. (1983) along with the categories and subcategories listed in Battig and Montague (1969) to create 770 sentence pairs. Similar to Ettinger (2019), we did not use multi-word category names.

**NEG-1500-SIMP-GEN**   We use human-in-the-loop and few-shot prompted GPT3-text-davinci-002 with default parameters, to generate 750 pairs (1500 sentences). Each GPT3 prompt consists of an instruction and four randomly selected in-context examples. The cost of generating the dataset is around $30. An example of the prompt is shown in the appendix A. After generating samples from GPT3, the dataset was manually cleaned to filter out poorly generated examples. Details about manual filtering in mentioned in Appendix B.1.

We analyzed the word distribution for the extended datasets and found that for NEG-1500-SIMP-GEN, few categories are more frequent than others, for example, the highest frequent category "animal" has three times more examples than the

---

[2]We use `text-davinci-002` version of GPT3

[3]Fischler et al. (1983) refers to these as "subjects" and "objects", respectively.

lowest frequent category "tree". This difference is 1.5 times in NEG-1500-SIMP-TEMP. Appendix D show the top 20 categories for all datasets.

## 3.2 Role-reversal Dataset: ROLE-1500

The original ROLE-88 dataset (Ettinger, 2019; Chow et al., 2016) is a role reversal dataset consisting 88 sentences (44 sentence pairs). Here is an example of a sentence pair: *"The librarian documented which journalist the celebrities had avoided." "The librarian documented which celebrities the journalist had interviewed"*. The first sentence has two words representing roles (here: journalist, celebrities), which are reversed in the second sentence. The target verb (the last verb in the sentence) is always preceded by the auxiliary "had". This dataset was created to observe the effect of role reversal on the target verb. We notice that all of the role words represent human roles (chef, worker, etc.) or animals (whale, dog, etc.). There were some pairs where the role reversal didn't change the target word, but they were semantically correct.

To extend ROLE dataset, we used the same method as NEG-1500-SIMP-GEN and generated 1500 sentences with temperature set to 0.64 [4]. We call this new dataset as ROLE-1500 and cost of generating it is $25. After generating samples from GPT3, the dataset was manually cleaned. An example of the prompt, details on data filtering and word distribution for ROLE-1500 is provided in the Appendix A, B.2, and D respectively. Word distribution analysis in ROLE-1500 revealed a threefold difference between the highest and lowest frequency categories, similar to the findings in NEG-1500-SIMP-GEN.

## 4 Evaluation on Extended Datasets

Following methodology from Lialin et al. (2022), we evaluated 22 models on the newly created negation and role reversal data (Figure 1). Both negation and role reversal tasks were first converted into a masked language modeling task, where the category (the final verb) was replaced with a mask token. For GPT2 and GPT3, the task was converted into a causal language modeling task, where the target was removed, and the model had to predict the target word. The responses of the models were evaluated against the true target.

For GPT3, we also performed zero-shot evaluation with two prompts: with and without instructions pre-pended to the input sentence (see Appendix A). We used the GPT3-text-davinci-002 model via OpenAI API with the default parameters with max_tokens as 1. GPT3 response was evaluated against gold label.

Our evaluation metric is the top-5 prediction accuracy, following Ettinger (2019). It is the percentage of the model responses where the gold label is among the top five predictions from the model. Table 1 shows the results for the original and extended datasets. We also included top 1, 10, and 20 accuracies in Appendix F. For the negation dataset, we used model sensitivity as an additional metric. Since we didn't have N400 amplitude and cloze probability for the extended datasets, we defined sensitivity as the percentage of sentence pairs for which the top-1 prediction changed, when "not" was added. Table 1 shows the sensitivity result. Results for all models are shown in Appendix E. We could not compute ROLE-1500 sensitivity, as in some sentence pairs the target word was the same.

To assess the reliability of the new datasets, we employed the methodology proposed by Card et al. (2020) to evaluate the statistical power of the datasets. Specifically, we used McNemar's test to assess the top two models on each dataset. For the ROLE dataset, our primary criterion was accuracy, while for the NEG dataset, we used the accuracy on affirmative examples (as in Ettinger (2019)). Additionally, we conducted a permutation test to check the statistical significance of the differences for extended datasets. The results are discussed in Section 5.

We validated our new datasets through human evaluation. For this, we randomly selected 100 samples from each of the extended datasets. Each of these sentences was presented to two annotators, who were asked to complete the sentences with a one-word response. The completion was compared against target labels. This is a method analogous to the cloze test used traditionally to gauge human language comprehension. We used Cohen's kappa to compute inter-annotator agreement.

## 5 Results

**Models performance dropped substantially on extended datasets.** Model performance[5]

---

[4]We set Hyperparameters by manually evaluating the output and selecting values that produced samples close to the specifications mentioned in the original dataset papers.

[5]Difference in the model performance is mentioned as the difference in percentage points. For example, if model A

| | ROLE-88 | NEG-136 SIMP(Aff) | NEG-136 Sensitivity | ROLE-1500 | NEG-1500 SIMP TEMP(Aff) | NEG-1500 SIMP GEN(Aff) | NEG-1500 SIMP-TEMP Sensitivity | NEG-1500 SIMP-GEN Sensitivity |
|---|---|---|---|---|---|---|---|---|
| $BERT_{base}$ | 27.3 | **100.0** | 16.7 | 20.3 | 58.4 | 55.3 | 53.5 | 35.9 |
| $BERT_{large}$ | 37.5 | **100.0** | 33.3 | 21.5 | 65.1 | 53.8 | 53.5 | 40.3 |
| $RoBERTa_{base}$ | 46.6 | 94.4 | 66.7 | 23.0 | 62.1 | 44.0 | 64.4 | 71.5 |
| $RoBERTa_{large}$ | **55.7** | 94.4 | 50.0 | 26.1 | 63.4 | 53.7 | 64.5 | 69.5 |
| $DistilBERT_{base}$ | 28.4 | 94.4 | 27.8 | 19.3 | 57.3 | 52.8 | 44.7 | 41.5 |
| $AlBERTv2_{base}$ | 26.1 | 33.3 | 38.9 | 15.3 | 10.0 | 11.7 | 37.3 | 35.9 |
| $AlBERTv2_{xlarge}$ | 37.5 | 94.4 | 27.8 | 21.1 | 40.4 | 47.8 | 52.7 | 51.1 |
| $AlBERTv2_{xxlarge}$ | 50.0 | **100.0** | 44.4 | **29.0** | 42.9 | 45.3 | 55.7 | 61.5 |
| $T5_{large}$ | 36.4 | 94.4 | 50.0 | 18.2 | 60.4 | 49.8 | 44.0 | 0.0 |
| $T5_{xl}$ | 44.3 | 83.3 | 66.7 | 21.5 | 60.9 | 60.9 | 68.2 | 70.0 |
| $GPT2_{base}$ | 0.0 | 0.0 | 66.7 | 11.2 | 48.3 | 37.7 | 60.4 | 56.4 |
| $GPT2_{xl}$ | 0.0 | 0.0 | 44.4 | 18.8 | 63.6 | 52.8 | 59.0 | 71.9 |
| GPT3 | 44.4 | 94.4 | **100.0** | 24.6 | **65.9** | **63.3** | **100.0** | **100.0** |
| $GPT3_{prompt}$ | 38.6 | 72.2 | **100.0** | 24.4 | 55.9 | 55.2 | **100.0** | **100.0** |

Table 1: Zero-shot top-5 word prediction accuracy and sensitivity (top-5 over the whole vocabulary). ROLE-88 and NEG-136 are from Lialin et al. (2022). ROLE-1500 and NEG-1500 are the new datasets. The best result for each task is in bold. "SIMP" stands for simple, "prompt" stands for prompting(adding instruction). The negation task is evaluated in the affirmative form (*Aff*). Sensitivity is the percentage of sentence pairs for which the top-1 prediction changed. See Appendix E for accuracy and sensitivity of all models on extended datasets.

dropped by 40-50% on negation (affirmative-only) and by 20-30% on role reversal for all models except GPT2. Even top-20 prediction accuracy couldn't reach the performance shown for the original datasets. For NEG-1500-SIMP-TEMP, BERT[6], RoBERTa, DistilBERT and T5 accuracy decreased 30% each, while ALBERT-xxlarge-v2 had the biggest drop of 57%. Compared to NEG-1500-SIMP-TEMP, NEG-1500-SIMP-GEN shows 5-10% more (absolute) drop in the performance. In ROLE-1500, the performance of BERT, DistilBERT and T5 decreased by about 5-15% each. RoBERTa and ALBERT showed the highest drop of 23-29% on this task. Increasing dataset size had a larger effect on model accuracy for negation than for the role-reversal task. The models that performed best on the small datasets did not remain in the lead on the extended. For example, in ROLE, RoBERTa-large was the best model on the original dataset, but ALBERT-v2-xxlarge surpassed it on the extended one. For negation, BERT and AL-BERT were best originally, but GPT3 surpassed them when the datasets became larger.

**GPT3 cannot solve, but can generate such examples.** While GPT3 is unable to reliably solve

these tasks, it is perfectly capable of picking up the pattern and generating valid examples. ROLE-1500 was generated by GPT3, and yet, RoBERTa-large, and ALBERT-xxlarge perform better on it than GPT3. There is a gap of around 4.5% between the best model and GPT3. Adding instructions to the prompt, didn't change the performance on ROLE-1500, but decreased the accuracy by 8-10% on negation. Sensitivity to negation remains the same for GPT3 with and without instruction prompts.

**GPT3 responses contain variety of mistakes.** Approximately 2/3 of the generated ROLE data was manually filtered out. 14.3% of responses were single sentences without a pair. 7.4% responses were semantically incorrect. Similar kind of errors were filtered out in the negation data as well. GPT3 also generated duplicate entries. 62.5% of negation data generated with GPT3 were duplicate samples, compared to 32.5% for the role reversal dataset.

**GPT3 generates more diverse examples than the template-based method.** While GPT-3 provided a way to generate diverse samples, the template-based method ensured consistency and alignment with the original dataset. Our result shows that GPT-3 generated samples offered a variety with 63 unique target labels whereas the template-based method provided controlled variability based on predefined structures with 23 unique target labels. 11 of these target labels are common between both

scores 40% and model B scores 20%, model B is worse by 20 percentage point and not by 50%.

[6]When we mention a model without specifying its size, we refer to the average percentage change for different sizes within that model family used in our experiments. For example, BERT performance will include BERT-base and BERT-large accuracies.

the datasets. The models exhibit similar performance when evaluated on datasets produced by both template as well as model based methods.

**Models are more sensitive to negation for extended dataset.** All models except ALBERT-v2-base, T5-large, and GPT2-base show higher sensitivity to negation on both extended negation datasets compared to the original one. For NEG-1500-SIMP-TEMP, the sensitivity for BERT and ALBERT-xl increased by 36.8% and 21-25%, respectively. For RoBERTa-base, T5-large and GPT2, sensitivity to negation dropped by 2.3-6%. The sensitivity to negation on NEG-1500-SIMP-GEN either increased or remained the same compared to original data for all models, except ALBERT-v2-base, T5-large and GPT2-base. For BERT and RoBERTa the sensitivity increased by almost 20%. Results demonstrate high levels of negation sensitivity in models like BERT and ALBERT, suggesting that previous findings might have been skewed due to smaller test sets.

**Change in model performance on extended datasets depends on its architecture and size.** Encoder-only models had the highest performance drop of 15% for ROLE-1500 and 40% for NEG-1500 (average of template-based and generated negation dataset). In comparison, the seq-to-seq models accuracy decreased by 14% in ROLE-1500 and 29% in NEG-1500, while decoder-only models show a gain of 5% in ROLE-1500 and 27% in NEG-1500. Encoder-only models also demonstrate the highest increase in negative sensitivity, with a gain of 13%, while seq-to-seq and decoder-only models gains only 1.2% and 7.4% respectively. When analyzed across model size, we found models of size around 60M show the highest drop of around 35% in NEG-1500, while models of size close to 300M experience the highest decrease of 14.6% in the ROLE dataset. The 300M size models exhibit the highest gain of 44% in negativity sensitivity (see tables in Appendix G).

**New extended datasets are more reliable than original small datasets.** The power analysis using McNemar's test reveals very low power for original datasets. ROLE-88 has a power of 0.26 whereas NEG-136's power is 0.01. For the extended dataset ROLE-1500, the power to differentiate between the top two accuracy models significantly improved to 0.72, and for GPT3-generated NEG-1500-SIMP, the power reached a full 1.00

in distinguishing top models. However, for the template-generated NEG-1500-SIMP, the power remained low: 0.04 for the top two models and 0.23 for the top 1 versus top 3. From these results, it is evident that ROLE-1500 and NEG-1500-SIMP-GEN are dramatically more reliable datasets. Specifically, they can distinguish between small effects of approximately 0.03 for ROLE and about 0.15 for NEG-1500-SIMP-GEN (minimum detectable effect, MDE).

The permutation test shows a statistically significant difference between the accuracies of the top-2 models for ROLE-1500 and NEG-1500-SIMP-GEN at a 0.05 significance level. P-values are 0.01 and 0.0002 respectively. In contrast, template generated dataset (NEG-1500-SIMP-TEMP) does not show statistically significant accuracies and fails the test with a p-value of 0.22.

**Human performance surpasses all model performance.** From human cloze test on ROLE-1500 and NEG-1500-SIMP(TEMP and GEN), we observed that human performance consistently exceeded model performance across all datasets. Specifically, for NEG-1500-SIMP-TEMP, human evaluation resulted in a top-1 accuracy of 58.5%, whereas the top-performing model, T5-xl, achieved about 40%. In the case of NEG-1500-SIMP-GEN, the human evaluation yielded an accuracy of 45.5%, 5.8% higher than the leading model, T5-xl. For ROLE, human performance (10%) is better than all of the model performances except ALBERT-xxlarge (v1 and v2). Human evaluation results are included in Tables 3, 4 and 5. Inter-annotator agreement for ROLE-1500 is 0.07, whereas for NEG-1500-SIMP-TEMP and NEG-1500-SIMP-GEN, it is 0.50 and 0.45 respectively.

# 6 Conclusion

We provide a new resource that dramatically extends existing probing data for negation and role reversal. The psycholinguistic datasets NEG-136-SIMP and ROLE-88 are extended to more reliable and large datasets, each with 1500 data points. We evaluate 22 models using the new data, observing that the absolute accuracy drops for most models on both tasks. However, most models do show an increased sensitivity to negation. Strikingly, as seen in the role reversal task, we show that GPT3 may be capable of generating the data for a task while failing to show top performance on it.

## Limitations

The current work has several limitations. The study did not include N400 amplitude. Another limitation is the smaller size of the original datasets. As we had a limited number of in-context samples that were drawn from original datasets, there was a limitation on the number of new data points we could generate. Additionally, the cost of generating samples with GPT3 was another limiting factor. Lastly, the frequency distribution of the categories in the new datasets is not balanced, which may introduce some bias in model evaluation. This imbalance exists in the original datasets too.

## Ethics Statement

Using large closed-source language models to generate evaluation data, while highly effective, should be approached with caution. First, neural networks of hundreds of billions of parameters are computationally expensive to infer and potential carbon dioxide emissions should be taken into consideration. Second, while API access to such models allows to avoid many technical challenges, especially having enough resources to inference large models, it comes with limitations. For example, GPT3 can only return the logprops of at most five top tokens over the vocabulary. Also, the API provides no control over the random seed used for sampling reducing the potential reprehensibility of the research. Finally, without the direct access to the model weights, there is no guarantee that this version of the model will be always accessible to researchers in the future and won't be restricted, modified, deleted, or lost.

## Acknowledgment

This project is funded in part by an NSF CAREER award to Anna Rumshisky (IIS-1652742). We would like to express our gratitude to Vijeta Deshpande and Sherin Muckatira for participating in human evaluation.

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

## A GPT3 prompt

### A.1 ROLE-1500 generation prompt

An example prompt for generating ROLE-1500 dataset is given below. The roles are colored orange and blue, whereas the target word is highlighted in green:

*The task is to reverse the role in the sentences. Generate more sentence like this:*

*The journalist investigated which athlete the team had recruited, The journalist investigated which team the athlete had joined,*

*The detective interviewed which witness the doctor had suspected, The detective interviewed which doctor the witness had seen.*

*The teacher lectured which student the class had ignored, The teacher lectured which class the student had left,*

*The police reported which criminal the witness had described, The police reported which witness the criminal had robbed,*

*The doctor treated which patient the nurse had examined, The doctor treated which nurse the patient had injured,*

### A.2 NEG-1500-SIMP-GEN generation prompt

An example prompt for generating NEG-1500-SIMP-GEN dataset is given below.

*"The task is to generate affirmative sentences and its negation. The object of the sentence should be a hypernym of the subject of the sentence. Generate more sentence pairs like these: "*

*A robin is a bird, A robin is not a tree,*
*An oak is a tree, An oak is not a flower,*
*A carrot is a vegetable, A carrot is not a bird,*
*A hammer is a tool, A hammer is not a tree,"*

### A.3 ROLE-1500 evaluation prompts with and without instructions

Prompt with instruction for ROLE-1500:

*"The goal is to complete the given sentence with one English word. Avoid punctuation or a new line or space. The librarian documented which journalist the celebrities had".*

The prompt without instruction: *"The librarian documented which journalist the celebrities had".* The model predicts the next word in sentence.

Instruction is same for ROLE-1500 and NEG-1500-SIMP-GEN. Only the in-context example changes.

## B Manual cleaning of the datasets

### B.1 NEG-1500-SIMP

For NEG-1500-SIMP-GEN, to get 1500 cleaned samples, we generated 12000 sentences. 87.5% of the responses were rejected as they were a) duplicates (62.5%) b) empty lines (15%) c) only negated sentences (8.75%) d) others (1.25%). Others include sentences like "A toy is a toy" and "A toy is an object", as well as objects with multiple words and semantically wrong sentences (e.g., "A flower is a rose", "A cloud is a weather"). The cost of generating NEG-SIMP-1500-GEN was $35.

### B.2 ROLE-1500

After generating GPT3 responses to these prompts, the samples were manually cleaned to get 1500 sentences (750 sentence pairs). We generated 4700 sentences (including empty, partial and no-pair sentences), out of which 68% were removed, as a) 32.5% were duplicates, b) 14.3% missed on getting a pair or were simple sentences, c) 8.5% were empty lines, d) 7.4% were semantically wrong sentences or had a wrong role reversal (e.g., "The camper reported which book the girl had eaten"), e) 5.3% were repeating instructions or partial sentences. In some instances, generated examples included some repetition of the role nouns present in the in-context examples.

## C Rules to create NEG-1500-SIMP-GEN

An affirmative sentence and a negation sentence make a pair. Affirmative sentences follow the format of "subject is an object" whereas its negation form is "subject is not an object". Affirmative sentences can be, for example *"Robin is a bird"* or *"Robin is an animal"*, the subject word is *"Robin"*, the object word is *"bird"* and its immediate superordinate category name is *"animal"*. The negation for any of these two affirmative sentences can be *"Robin is not a plant"* or *"Robin is not a tree"*. The target completion ( e.g. *"plant" or "tree"*) has to be an object or superordinate word from another category. Figure 1 in the Appendix 1 shows two categories (Animal and Plant) together with its respective subject and object words (Battig and Montague, 1969).

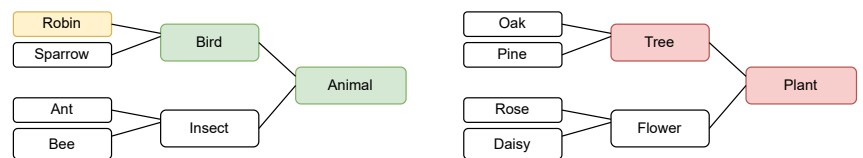

Figure 1: Sentence subject word (Robin). Affirmative sentence completion are in green - object word(Bird) and super ordinate category name(Animal). Negative sentence completion are in red - object word(Tree) and super ordinate category name(Plant). (Battig and Montague, 1969)

## D  Top 20 target words for all datasets

Figure 2 and 6 show top 20 object words for original datasets NEG-136-SIMP and ROLE-88. Figure 3, 4 and 5 depict top 20 object words for original new extended datasets. Y axis represents the number of times the target words. NEG-136-SIMP has less than 20 categories, therefore all of them has been shown in the figure 2.

## E  Result for all models

Table 2 shows the top five accuracy for all the models.

## F  Top 1, 10 and 20 prediction accuracy for all models

Table 4 and 3 show top 1, 10 and 20 accuracy for new negation datasets. Table 5 depict top 1, 10 and 20 accuracy for new role reversal dataset.

## G  Results across model type and model size

Table 6 and 7 show the change in the model accuracy for extended dataset as compared to the original dataset across model type and model size respectively. NEG-1500 represents the average of NEG-1500-SIMP-TEMP and NEG-1500-SIMP-GEN. We didn't mention 1B and 175B model sensitivity in Table 7 as they were 100% in original and extended datasets, therefore the difference is zero.

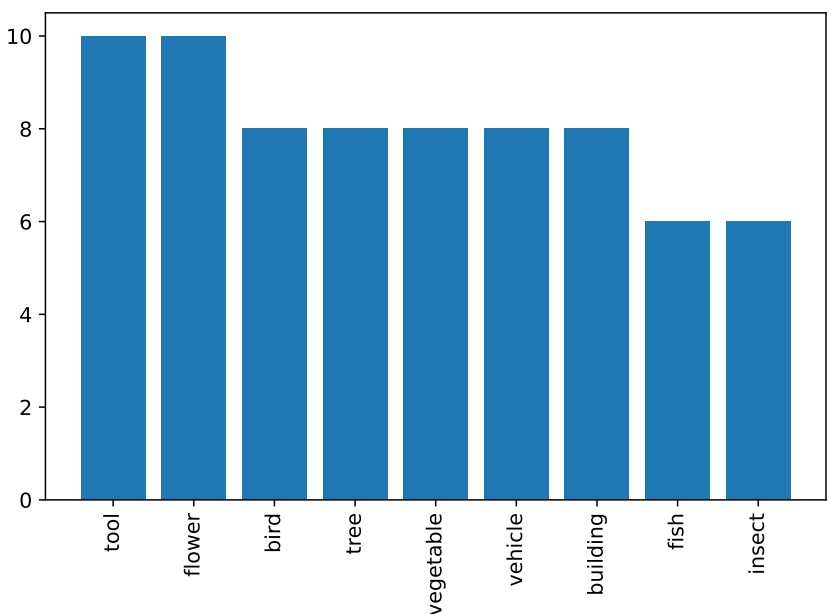

Figure 2: Top categories for NEG-136-SIMP

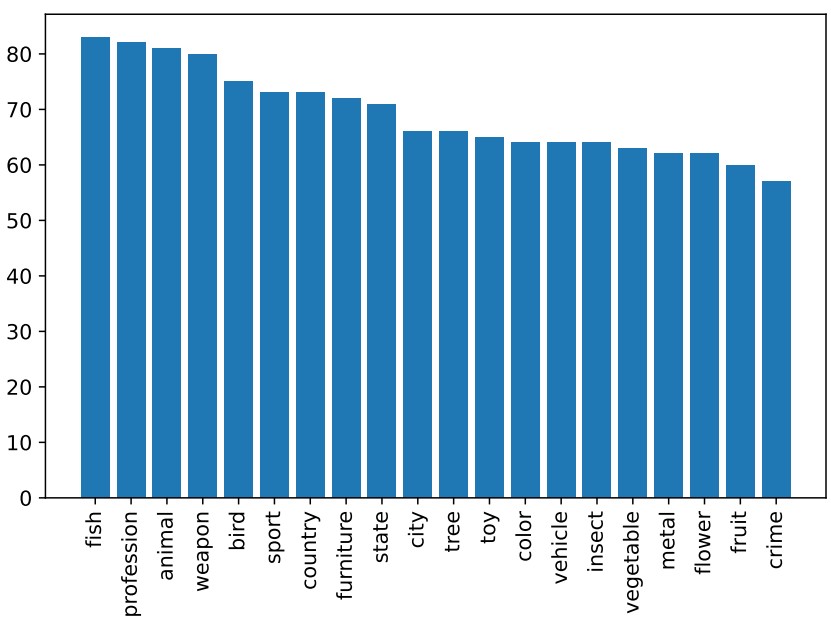

Figure 3: Top 20 categories for NEG-1500-SIMP-TEMP

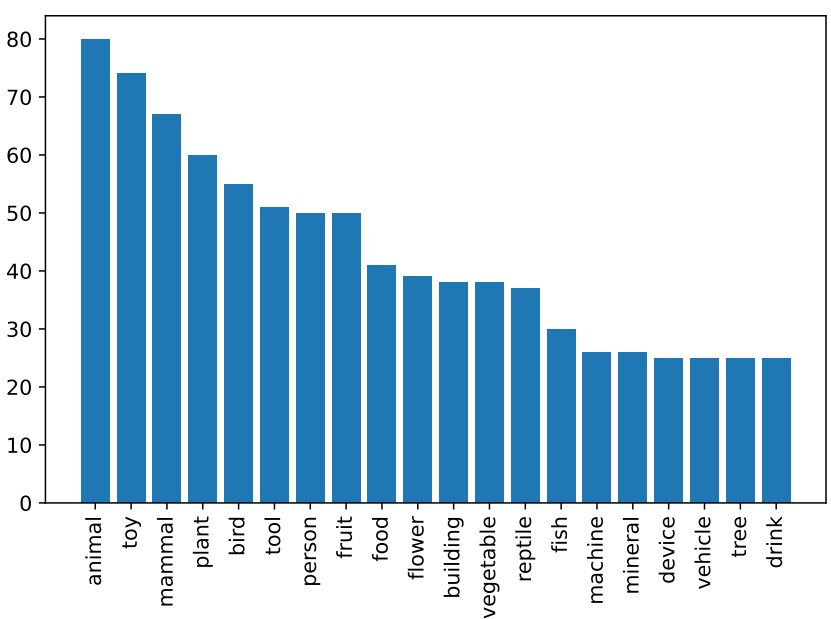

Figure 4: Top 20 categories for NEG-1500-SIMP-GEN

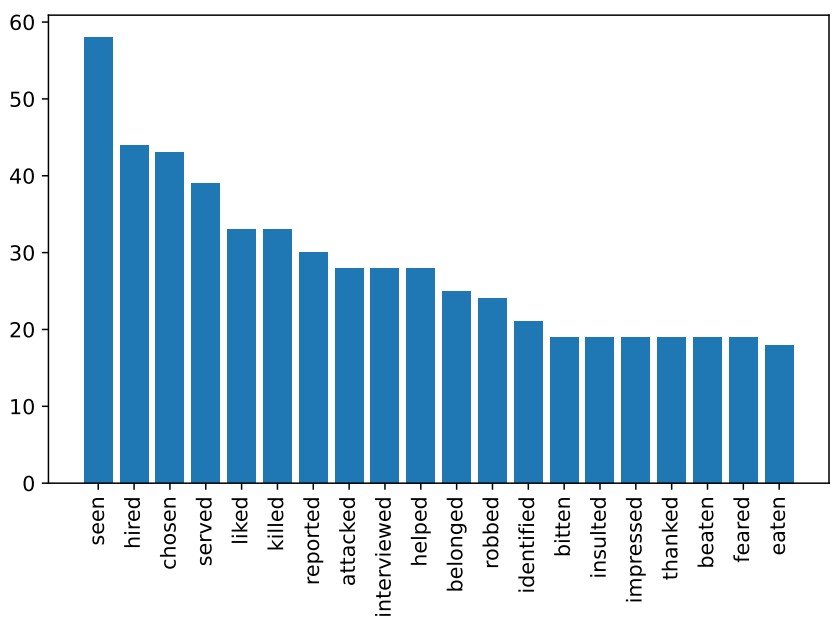

Figure 5: Top 20 categories for ROLE-1500

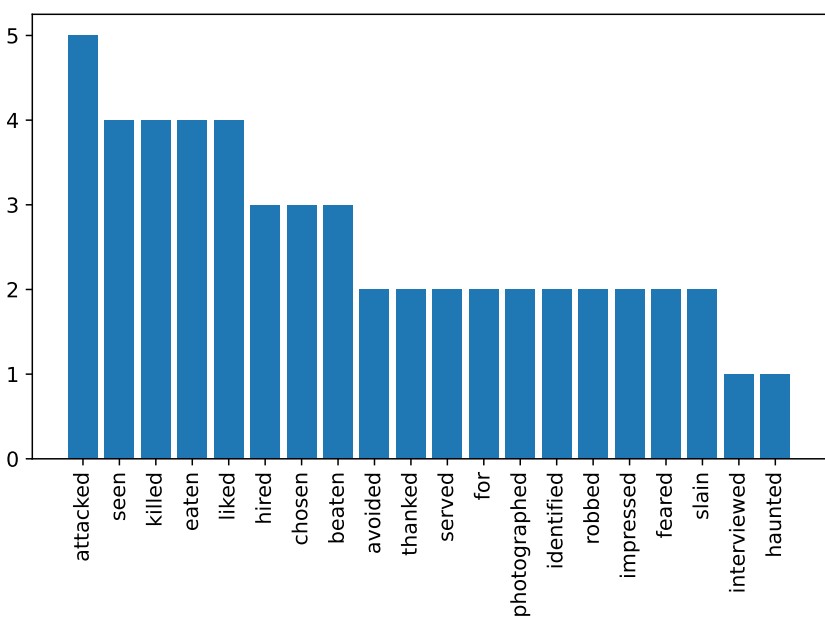

Figure 6: Top 20 target words for ROLE-88

| | ROLE-88 | NEG-136 SIMP(Aff) | NEG-136 Sensitivity | ROLE-1500 | NEG-1500 SIMP TEMP(Aff) | NEG-1500 SIMP GEN(Aff) | NEG-1500 SIMP-TEMP Sensitivity | NEG-1500 SIMP-GEN Sensitivity |
|---|---|---|---|---|---|---|---|---|
| BERT$_{base}$ | 27.3 | **100.0** | 16.7 | 20.3 | 58.4 | 55.3 | 53.5 | 35.9 |
| BERT$_{large}$ | 37.5 | **100.0** | 33.3 | 21.5 | 65.1 | 53.8 | 53.5 | 40.3 |
| RoBERTa$_{base}$ | 46.6 | 94.4 | 66.7 | 23.0 | 62.1 | 44.0 | 64.4 | 71.5 |
| RoBERTa$_{large}$ | **55.7** | 94.4 | 50.0 | 26.1 | 63.4 | 53.7 | 64.5 | 69.5 |
| DistilBERT$_{base}$ | 28.4 | 94.4 | 27.8 | 19.3 | 57.3 | 52.8 | 44.7 | 41.5 |
| AlBERTv1$_{base}$ | 17.1 | 72.2 | 22.2 | 10.4 | 37.1 | 36.4 | 40.0 | 35.6 |
| AlBERTv1$_{large}$ | 26.1 | 83.3 | 22.2 | 17.4 | 48.4 | 42.4 | 38.6 | 32.9 |
| AlBERTv1$_{xlarge}$ | 34.1 | 55.5 | 55.6 | 19.5 | 22.3 | 26.0 | 77.0 | 62.5 |
| AlBERTv1$_{xxlarge}$ | 53.4 | 72.2 | 55.6 | 28.5 | 38.7 | 39.3 | 59.9 | 61.7 |
| AlBERTv2$_{base}$ | 26.1 | 33.3 | 38.9 | 15.3 | 10.0 | 11.7 | 37.3 | 35.9 |
| AlBERTv2$_{large}$ | 29.5 | 83.3 | 22.2 | 18.8 | 36.6 | 36.2 | 31.7 | 33.1 |
| AlBERTv2$_{xlarge}$ | 37.5 | 94.4 | 27.8 | 21.1 | 40.4 | 47.8 | 52.7 | 51.1 |
| AlBERTv2$_{xxlarge}$ | 50 | **100.0** | 44.4 | **29.0** | 42.9 | 45.3 | 55.7 | 61.5 |
| T5$_{small}$ | 9.1 | 44.4 | 0.0 | 5.8 | 33.6 | 25.3 | 16.8 | 19.6 |
| T5$_{base}$ | 27.3 | 88.9 | 27.8 | 14.8 | 58.2 | 45.1 | 46.5 | 33.5 |
| T5$_{large}$ | 36.4 | 94.4 | 50.0 | 18.2 | 60.4 | 49.8 | 44.0 | 0.0 |
| T5$_{xl}$ | 44.3 | 83.3 | 66.7 | 21.5 | 60.9 | 60.9 | 68.2 | 70.0 |
| GPT2$_{base}$ | 0.0 | 0.0 | 66.7 | 11.2 | 48.3 | 37.7 | 60.4 | 56.4 |
| GPT2$_{medium}$ | 0.0 | 0.0 | 50.0 | 16.4 | 61.4 | 46.9 | 53.1 | 60.7 |
| GPT2$_{large}$ | 0.0 | 0.0 | 38.9 | 16.8 | 61.7 | 51.7 | 57.9 | 69.2 |
| GPT2$_{xl}$ | 0.0 | 0.0 | 44.4 | 18.8 | 63.6 | 52.8 | 59.0 | 71.9 |
| GPT3 | 44.4 | 94.4 | **100.0** | 24.6 | **65.9** | **63.3** | **100.0** | **100.0** |
| GPT3$_{prompt}$ | 38.6 | 72.2 | **100.0** | 24.4 | 55.9 | 55.2 | **100.0** | **100.0** |

Table 2: Zero-shot top-5 word prediction accuracy and sensitivity (top-5 over the whole vocabulary). ROLE-88 and NEG-136 are from Lialin et al. (2022). ROLE-1500 and NEG-1500 are the new datasets. The best result for each task is in bold. "SIMP" stands for simple, "prompt" stands for prompting. The negation task is evaluated in the affirmative form (*Aff*). Sensitivity is the percentage of sentence pairs for which the top-1 prediction changed.

| NEG-1500-SIMP-TEMP | Top 1 | Top 10 | Top 20 |
|---|---|---|---|
| BERT$_{base}$ | 28.5 | 64.7 | 71.4 |
| BERT$_{large}$ | 34.2 | **71.9** | **76.5** |
| RoBERTa$_{base}$ | 22.2 | 68.7 | 73.0 |
| RoBERTa$_{large}$ | 30 | 69.2 | 74.4 |
| DistilBERT$_{base}$ | 28.6 | 67.8 | 75.6 |
| AlBERTv1$_{base}$ | 14.3 | 43.9 | 49.5 |
| AlBERTv1$_{large}$ | 13.5 | 56.9 | 63.3 |
| AlBERTv1$_{xlarge}$ | 10.3 | 28.0 | 32.6 |
| AlBERTv1$_{xxlarge}$ | 12.1 | 48.6 | 57.3 |
| AlBERTv2$_{base}$ | 2.0 | 18.8 | 29.2 |
| AlBERTv2$_{large}$ | 10.6 | 46.8 | 55.6 |
| AlBERTv2$_{xlarge}$ | 24.7 | 45.2 | 50.0 |
| AlBERTv2$_{xxlarge}$ | 14.8 | 53.4 | 62.8 |
| T5$_{small}$ | 1.7 | 43.6 | 51.6 |
| T5$_{base}$ | 21.4 | 65.6 | 71.3 |
| T5$_{large}$ | 22.7 | 68.3 | 73.8 |
| T5$_{xl}$ | **39.7** | 66.2 | 69.9 |
| GPT2$_{base}$ | 0.0 | 57.1 | 63.6 |
| GPT2$_{medium}$ | 0.0 | 69.5 | 73.2 |
| GPT2$_{large}$ | 0.0 | 70.0 | 75.7 |
| GPT2$_{xl}$ | 0.0 | 70.1 | 75.6 |
| GPT3 | 27.5 | N/A | N/A |
| GPT3$_{prompt}$ | 21.8 | N/A | N/A |
| Human cloze completion | **58.5** | - | - |

Table 3: Zero-shot top-1,10 and 20 word prediction accuracy on NEG-1500-SIMP-TEMP. Top-5 is selected over the whole model vocabulary. The best result on each task is highlighted in bold. SIMP stands for simple, TEMP stands for template. Negation tasks are evaluated in the affirmative form (*aff*). GPT3 only produces top 5 predictions. Human cloze completion accuracy is the average accuracy of two annotators.

| NEG-1500-SIMP-GEN | Top 1 | Top 10 | Top 20 |
|---|---|---|---|
| BERT$_{base}$ | 18.8 | 66.0 | 73.3 |
| BERT$_{large}$ | 22.8 | 64.3 | 72.0 |
| RoBERTa$_{base}$ | 15.4 | 51.6 | 60.1 |
| RoBERTa$_{large}$ | 24.4 | 61.6 | 71.7 |
| DistilBERT$_{base}$ | 22.5 | 61.9 | 70.5 |
| AlBERTv1$_{base}$ | 11.2 | 46.0 | 53.6 |
| AlBERTv1$_{large}$ | 12.9 | 50.5 | 59.6 |
| AlBERTv1$_{xlarge}$ | 7.2 | 34.4 | 41.6 |
| AlBERTv1$_{xxlarge}$ | 12.4 | 50.7 | 60.7 |
| AlBERTv2$_{base}$ | 2.5 | 22.2 | 32.3 |
| AlBERTv2$_{large}$ | 10.7 | 44.8 | 54.7 |
| AlBERTv2$_{xlarge}$ | 24.0 | 56.5 | 64.5 |
| AlBERTv2$_{xxlarge}$ | 15.3 | 55.3 | 54.1 |
| T5$_{small}$ | 2.9 | 36.3 | 45.6 |
| T5$_{base}$ | 15.1 | 65.5 | 65.3 |
| T5$_{large}$ | 15.1 | 60.5 | 68.3 |
| T5$_{xl}$ | **39.7** | 66.2 | 69.9 |
| GPT2$_{base}$ | 16.4 | 48.0 | 60.0 |
| GPT2$_{medium}$ | 26.1 | 56.1 | 66.9 |
| GPT2$_{large}$ | 29.3 | 60.7 | 69.6 |
| GPT2$_{xl}$ | 30.4 | 60.1 | 70.4 |
| GPT3 | 26.5 | N/A | N/A |
| GPT3$_{prompt}$ | 24.3 | N/A | N/A |
| Human cloze completion | **45.5** | - | - |

Table 4: Zero-shot top-1,10 and 20 word prediction accuracy on NEG-1500-SIMP-GEN. Top-5 is selected over the whole model vocabulary. The best result on each task is highlighted in bold. SIMP stands for simple, negation tasks is evaluated in the affirmative form. GPT3 allow only till top 5 predictions.

| ROLE-1500 | Top 1 | Top 10 | Top 20 |
|---|---|---|---|
| BERT$_{base}$ | 6.7 | 28.1 | 38.9 |
| BERT$_{large}$ | 9.1 | 30.9 | 40.7 |
| RoBERTa$_{base}$ | 0 | 33.9 | 47.3 |
| RoBERTa$_{large}$ | 0 | **38.6** | **52** |
| DistilBERT$_{base}$ | 6.5 | 27.3 | 36.9 |
| AlBERTv1$_{base}$ | 2.8 | 16.2 | 25.3 |
| AlBERTv1$_{large}$ | 4.8 | 24.2 | 32.7 |
| AlBERTv1$_{xlarge}$ | 7.2 | 27.3 | 35.5 |
| AlBERTv1$_{xxlarge}$ | **12.2** | 37.2 | 45.1 |
| AlBERTv2$_{base}$ | 6.2 | 22.7 | 29.9 |
| AlBERTv2$_{large}$ | 5.9 | 27.0 | 36.1 |
| AlBERTv2$_{xlarge}$ | 8.8 | 29.6 | 37.9 |
| AlBERTv2$_{xxlarge}$ | 11.6 | 37.1 | 45.1 |
| T5$_{small}$ | 1.3 | 12.1 | 18.3 |
| T5$_{base}$ | 4.7 | 21.9 | 28.5 |
| T5$_{large}$ | 6.2 | 25.3 | 31.3 |
| T5$_{xl}$ | 7.9 | 29.0 | 36.5 |
| GPT2$_{base}$ | 0 | 18.9 | 27.1 |
| GPT2$_{medium}$ | 0 | 25.2 | 36.5 |
| GPT2$_{large}$ | 0 | 27.0 | 37.7 |
| GPT2$_{xl}$ | 0 | 29.9 | 38.8 |
| GPT3 | 7.7 | N/A | N/A |
| GPT3$_{prompt}$ | 5.7 | N/A | N/A |
| Human cloze completion | **10.0** | - | - |

Table 5: Zero-shot top-1,10 and 20 word prediction accuracy on ROLE-1500. Top-5 is selected over the whole model vocabulary. The best result on each task is highlighted in bold. GPT3 produces only top 5 predictions.

| Model type | ROLE-1500 | NEG-1500 | NEG-1500 sentivity |
|---|---|---|---|
| Encoder-Only | -15.32 | -39.52 | 13.07 |
| Seq-to-Seq | -14.20 | -28.48 | 1.20 |
| Decoder-Only | 4.87 | 27.60 | 7.38 |

Table 6: Change in the model accuracy for extended dataset as compared to the original dataset across model type. Negative sign shows drop in the model performance whereas positive number shows the gain in model performance when dataset was extended.

| Model type | ROLE-1500 | NEG-1500 | NEG-1500 sentivity |
|---|---|---|---|
| <60M | -10.04 | -34.83 | 13.6 |
| 100M | -6.47 | -13.83 | 6.98 |
| 300M | -14.60 | -24.77 | 44.52 |
| 1B | -1.35 | 13.3 | NA |
| 175B | -17.00 | -23.23 | NA |

Table 7: Change in the model accuracy for extended dataset as compared to the original dataset across model size. Negative sign shows drop in the model performance whereas positive number shows the gain in model performance when dataset was extended.