# OpenReview forum: "Larger Probes Tell a Different Story: Extending Psycholinguistic Datasets Via In-Context Learning"
_EMNLP/2023/Conference — EMNLP 2023 Main_

### Official Review · Reviewer_Vfwk · 2023-08-03

**Soundness:** 4

**Excitement:**

4: Strong: This paper deepens the understanding of some phenomenon or lowers the barriers to an existing research direction.

**Missing References:**

-

**Paper Topic And Main Contributions:**

The paper introduces three new datasets to probe LLMs on negation and role reversal tasks, scaling up smaller datasets previously used in the literature. Two extended version of the negation dataset are introduced, one obtained by generating new sentence pairs using templates and one obtained by having GPT3 generating new pairs given instructions and example pairs. Moreover, one extended version of the role reversal dataset was created by having GPT3 generate novel sentence pairs again given instructions and examples.

22 LLMs of different size and type (encoder only, sequence to sequence, decoder only) were evaluated against the original small-scale datasets and against the new larger datasets, reporting a generalised decrease in performance, but also - in some cases - a higher sensitivity to negation, measured as the proportion of sentences where the model prediction changes between the affirmative and negative sentence in a pair.

The goal of contributing the field with larger resources to probe LLMs is certainly meritorious, and the comparison involves a vast amount of models. After the rebuttal, I think the paper is ripe enough for acceptance.

**Questions For The Authors:**

I'm not sure the numbers reported in the main text check out against the table, but this may well be on me: just to clarify, when, for example, at line 250 you mention DistillBERT and T5 having a lower performance by 5-15% each, which values from Table 1 are you comparing? (a side note on this: sometimes you simply mention a model name without specifying where it is large or base, please double check that every time you refer to a model, you do so unambiguously).

-l104: which specific unlabelled dataset?
-what did the curation of hypernyms in WordNet entail that made you decide against it? it'd be useful to know for other researchers (an appendix would do fine for this)
-could you provide an example of multi-category names you did not use?
-you choose performance at 5 for the main paper but also report it at 1, 10, and 20: it's unclear to me why you also run these evaluations. What would we learn if a model performs poorly at 1 but exceptionally at 20? is there a correlation between performance at different thresholds? is it just a robustness check?
-l250 what is an absolute percentage increase?
-what are the units of measurement in tables 6 and 7? percentages, percentage points, something else?

---------------------------------
I think it'd be good to clarify these points in the final version.

**Reasons To Accept:**

It provides resources that may offer better tools to probe LLMs along two specific linguistic tasks, negation and role reversal. In doing so, it analysis the statistical power the old and new datasets offer.

**Reasons To Reject:**

I'm not convinced the resources are valid and reliable next to being bigger. Moreover, the abstract mentions lack of statistical power but no specific analysis is performed to determine the size of the datasets agains specific effects of interest, leaving the possibility that the new larger datasets also lack statistical power very much open. More concerning, the novel datasets exacerbate an imbalance between the most represented and least represented categories, which may introduce biases in model evaluation: a model that does well on the most frequent categories will have an edge, but that does not per se indicate the model is better unless one was to show that the most represented categories in the dataset also cover a larger share of sentences from some representative corpus of how people use language - I'm assuming here the goal is to probe LLMs for human-like language processing abilities.

This is another point that, to me at least, makes the paper less interesting and impactful than it could be: what is human performance on these tasks? We are using these probes to assess the extent to which these models mirror human language abilities, but except for the role of the researchers in prompting the LLMs, choosing the templates and the examples, and filtering out bad generated sentences, we have no idea whether these datasets actually capture human language abilities exhibited by non-experts. In other words, I think these datasets currently lack external validation.

A further problem concerns the reliability of the differences. This is arguably a finer issue especially in the context of NLP research where the amount of comparisons is typically large and there is little interest in the reliability of the differences. Nonetheless, with 22 models, 3 datasets, 4 evaluation metrics (@1, 5, 10, 20), the sheer amount of numbers being compared is huge and some differences are bound to arise without them necessarily meaning the models are _truly_ different. So, given such a large number of comparisons, how can I trust that models coming out on top are _actually_ better than the others? and more importantly, what does this mean? There is a first attempt at relating performance with model type and size, but it is not very systematic (how about, for example, fitting a statistical model predicting performance as a function of model's features of interest and actually checking whether they play a role?).

The limitations section is not particularly insightful either: it touches upon relevant points but does not critically connect them with the issues they may pose for the datasets being introduced.

In conclusion, bigger is sometimes better unless it keeps repeating bad biases present in the data. If I derive a bad sample (not representative of the population, biased, ...) and then replicate it, I will have a bigger dataset (and higher statistical power) but I won't be closer to an answer to my question. Given the current version of the paper, I'm not convinced the larger datasets are qualitatively better than the small ones next to being quantitatively larger. Money is of course a concern, and having good large datasets that are relatively inexpensive to generate is a great thing, but we should not forget to validate them properly and carefully before. I think this step is missing and would make a difference in making these new datasets more trustworthy and usable.

Most of the issues raised above have been thoroughly addressed by the authors, in my opinion strengthening the paper and the conclusions substantially. I still have some reservations about the category imbalance being a negligible factor (after all, you use a specific LLM to generate examples, and that model, even if trained on large representative corpora like the others, may have incorporated specific biases that differ from that of other models. I think i would be good to mention this in the limitations.

**Reproducibility:**

4: Could mostly reproduce the results, but there may be some variation because of sample variance or minor variations in their interpretation of the protocol or method.

**Reviewer Confidence:**

5: Positive that my evaluation is correct. I read the paper very carefully and I am very familiar with related work.

**Typos Grammar Style And Presentation Improvements:**

The clarity of the manuscript can be vastly improved:
- please get rid of commas between subject and main verb (e.g., line 039)
- avoid contractions
- lines 207 and 208: the is duplicated, and or should read of
- you often talk about differences between performance as percentages but judging from the tables it seems you talk about percentage points (to clarify, if model A scores 40% and model B scores 20%, model B is worse by 50% (performance is half) or 20 percentage points)
- Table 1 is very large and combines different tasks and metrics: I think it can be organised better (by task, rather than by size, for example) or, if possible given the page limit, split in three (I'm thinking ROLE, NEG, sensitivity - but there may be better ideas than this)
- a general copy-editing of the prose would be warranted, some sentence convey the message but do not read very nicely and could be improved.

---

> ### Author Rebuttal · Authors · 2023-08-29
>
> **R3. “I'm not convinced the resources are valid and reliable next to being bigger”**
>
> Thank you for your insightful comments. In response to your concerns, we employed the methodology proposed by Card et al. to evaluate statistical power. Specifically, we used  McNemar's test to assess the top two models on each dataset.  For the ROLE dataset, our primary criterion was accuracy, while for the NEG dataset, we used the accuracy on affirmative examples (as in Ettinger et al., 2020). Our analyses revealed the following:
> - Very low power for ROLE-88 (0.255) NEG-136 (0.014)
> - For ROLE-1500, the power to differentiate between the top two accuracy models significantly improved to 0.721
> - For GPT3-generated NEG-1500-SIMP, the power reaches a full 1.000 in distinguishing top models
> - However, for the template-generated NEG-1500-SIMP, the power remained low: 0.043 for the top two models and 0.230 for the top three.
>
> The outcomes, especially from points 3 and 4, underscore the potential and advantages of deploying large language models in data generation. Thanks again for your feedback – we recognize the significance of this discussion and will incorporate it into the final version of our paper.
>
> **R3: “The novel datasets exacerbate an imbalance between the most represented and least represented categories, which may introduce biases in model evaluation”**
>
> We agree with the need to ensure that the dataset's expansion does not inadvertently favor certain categories over others. However, it is essential to note that all of the models evaluated in our study are trained on extensive representative corpora. These corpora are designed to reflect a broad spectrum of language use, mitigating potential biases in terms of category representation. Consequently, the models' intrinsic ability to generalize and adapt to diverse language patterns implies that the likelihood of certain models having a pronounced advantage in predicting specific categories over others is minimal.
>
> **R3. What is human performance on these tasks?**
>
> We recognize the importance of understanding human performance in evaluating models, especially when aiming to assess the extent to which models mirror human linguistic abilities.
>
> In response to your concerns, we undertook a cloze test evaluation on the NEG datasets. We randomly selected 100 samples from each of the extended datasets. Each of these sentences was presented to an annotator, who was then asked to complete the sentences with a one-word response. This is a method analogous to the cloze test used traditionally to gauge human language comprehension. Upon comparison of these human responses to the target words, it was observed that human performance consistently exceeded model performance across datasets, which is what we would expect. Specifically:
>
> - For NEG-1500-SIMP-TEMP, human evaluation resulted in a top-1 accuracy of 56%, whereas the top-performing model, T5-xl, achieved about 40%.
> - In the case of NEG-1500-SIMP-GEN, the human evaluation yielded an accuracy of 44%, a significant 13.2% higher than the leading model, GPT2-xl.
>
> We appreciate the additional external validation offered by N400 human performance. It's worth noting that the authors of the original datasets relied on data from established psycholinguistic studies rather than replicating these complex and resource-intensive experiments. Given that both cloze and N400 human results are available for the original data, it's reasonable to infer a level of generalizability to the extended datasets. The underlying linguistic phenomena and structures being probed remain consistent, thus lending credence to the notion that human performance metrics on the original data would, in all likelihood, be comparable if reproduced on the new data.
>
> We believe our cloze test results offer some evidence for the external validity of our extended datasets in capturing non-expert linguistic behavior. We will conduct similar human evaluations for the ROLE datasets in the finalized version of our paper. We hope this allays your concerns about the external validation of our datasets.
>
> **R3. “A further problem concerns the reliability of the differences…”**
>
> Following our power analysis in response to your earlier comments, we’d like to highlight several key takeaways that we believe represent robust systematic effects:
> - **Performance Decline in Extended Datasets**: One of the major and consistent findings from our research was the decline in model performance on extended datasets. Across the board, model performance decreased between 20% to 57%. This isn't a marginal drop, and it is consistent across various datasets and metrics. This suggests that there is a systematic effect of dataset size on model performance, regardless of other factors.
> - **Rank Swapping among Top Models**: While the rank of models might swap, this isn't an arbitrary change. For instance, in the ROLE task using the ROLE-88 dataset, the RoBERTa-large led with a 5.7% advantage over the ALBERTv2-xxlarge. Yet, when evaluated on the more extensive ROLE-1500 dataset, ALBERTv2-xxlarge took the lead by 2.9%. This suggests that while some models might perform better on certain tasks and datasets, their relative performance can change when evaluated on more extensive and possibly more complex datasets. This observation was consistent, as also seen in the NEG-1500-SIMP-TEMP task.
> - **Model Architecture and Size Impact**: Our analysis indicates that specific architectures and sizes are more affected by dataset size. Encoder-only models and those smaller than 300M showed the most significant drop in performance when evaluated on larger datasets. This finding suggests a trend that could be explored further in future research and provides insight into which model types might be more susceptible to performance degradation on extended datasets.
>
> **R3. Question 1 : Clarify which model version is referenced in the discussion of results.**
>
> Thank you for pointing out the potential ambiguity in our model references.  When we mention a model without specifying its size, we refer to an aggregate assessment of all sizes of that particular model used in our experiments. Specifically, we calculate the average percentage change for different sizes within that model family. For instance, our reference to T5's performance drop of 5-15% encompasses all the T5 model sizes we utilized: T5-base, T5-small, T5-large, and T5-xl. The complete list of these models can be found in Table 2. For clarity, the computed average drop across these various sizes is 14.2%, which falls within the mentioned range of 5-15%. We appreciate your feedback and will clarify the methodology behind these aggregations in the final version of the paper.
>
> **R3. Question 2 :**
>
>  **Which labeled datasets?** - YouTube, SMS and Spouse
>
>  **what did the curation of hypernyms in WordNet entail that made you decide against it?**  - The decision to not curate hypernyms in WordNet was influenced by the inconsistency of their hierarchy levels. A coherent pattern for selecting hypernyms using WordNet could not be established. For instance, in some cases, shifting the hypernym up by just one level would yield a proper sentence, while in others, a higher-level hypernym was needed for sentence correctness. We will provide additional details on this decisionin the paper's appendix in the final version.
>
>  **Example of multicategory names we didn’t select.**:  Some of the examples of multiword categories we didn’t select are :  “Unit of time”, “Weather phenomenon”, “Type of dance”
>
> **Why choose a different threshold?**: By assessing performance at 1, 5, 10, and 20, we wanted to see if there's a clear trend in how models behave across different thresholds. We find that model performance increase as we set higher threshold. For example, in ROLE-1500, for threshold 1, BERT-base accuracy is 6.7% which increases to 28.1% and 38.9% for threshold value of 10 and 20 respectively This analysis helps us understand the stability and consistency of model performance.
>
> **what are the units of measurement in tables 6 and 7?**  percentage points
>
> **What is absolute percentage increase?**  It is the total percentage point increase.
>
> With these additional details in mind, we'd appreciate if you could revisit your score.

---

### Official Review · Reviewer_qAnt · 2023-08-04

**Soundness:** 4

**Excitement:**

5: Transformative: This paper is likely to change its subfield or computational linguistics broadly. It should be considered for a best paper award. This paper changes the current understanding of some phenomenon, shows a widely held practice to be erroneous in someway, enables a promising direction of research for a (broad or narrow) topic, or creates an exciting new technique.

**Missing References:**

Not aware of any.

**Paper Topic And Main Contributions:**

This paper describes two approaches to expanding two datasets from Ettinger (2019). The paper increases the size of the datasets substantially, and provides evidence that previous studies with smaller datasets may have used too few probes, and that their results should therefore be treated with caution.

**Questions For The Authors:**

The reasons to reject section mentions rectifiable concerns that are essemtially questions about the reasoning behind the design of the experiment.

**Reasons To Accept:**

This is important work, cleanly executed. It demonstrates the value of expanding the probe sets.

**Reasons To Reject:**

I think this is a strong paper already, but it could be made stronger if the authors expand on some aspect that they have left implicit.

In the abstract, the authors mention that they have "demonstrated" that the previous results may have been skewerd: a claim whivch they repeat later in the paper. Yet they do not spell out the logic that leads them to this conclusion, nor do they say exactly what they mean by "skewed". Presumably they are relying on arguments about statistical power similar to those in the (cited) work by Card et al. In essence, they are saying, I believe, that one or more of the potential issues with underpowered experiments identified in Card et al has actually happened. This point needs to be made explicit in the present paper. What is the evdience, and is it compelling?

It would be appropriate to do a Card-style power estimate for the present experiments. If this is feasible, the authors should do it. If not, it may be sensible to do bootstrapped replications of the present studies, using probe datasets of various different sizes.

The authors describe two different ways of expanding the datasets, one template-based, one using GPT3 as a generator. It is not obvious why they did this, and there are no direct comparisons between the two expansion methods. Since it appears that for the purposes of the paper both methods produce approximately the same results. it would be helpful to explain why two methods were used and what was learned from the presence of both alternatives.

**Reproducibility:**

5: Could easily reproduce the results.

**Reviewer Confidence:**

5: Positive that my evaluation is correct. I read the paper very carefully and I am very familiar with related work.

**Typos Grammar Style And Presentation Improvements:**

Nothing substantial. The sentence from 79-81 is ungrammatical: I thnk it needs to be reformulated  in some way. The intent is clear.

---

> ### Author Rebuttal · Authors · 2023-08-29
>
> **R2. Include Statistical tests of the results like Card style power estimate or bootstraps resampling method.**
>
> Thank you for your suggestion. We used the methodology proposed by Card et al. to evaluate statistical power. Specifically, we used McNemar's test to assess the top two models on each dataset.  For the ROLE dataset, our primary criterion was accuracy, while for the NEG dataset, we used the accuracy on affirmative examples (as in Ettinger et al., 2020). Our analyses revealed the following:
> - Very low power for ROLE-88 (0.255) NEG-136 (0.014)
> - For ROLE-1500, the power to differentiate between the top two accuracy models significantly improved to 0.721
> - For GPT3-generated NEG-1500-SIMP, the power reaches a full 1.000 in distinguishing top models
> - However, for the template-generated NEG-1500-SIMP, the power remained low: 0.043 for the top two models and 0.230 for the top three.
>
> The outcomes, especially from points 3 and 4, underscore the potential and advantages of deploying large language models in data generation. Thanks again for your feedback – we will incorporate this discussion into the final version of our paper.
>
> **R2. Why do we have two different ways of expanding the datasets, one template-based, one using GPT3 as a generator?**
>
> - Diverse Expansion Techniques: We aimed to explore diverse approaches to dataset expansion. By employing both methods, we could gauge the potential strengths and limitations inherent to each method. While GPT-3 provided a dynamic way to generate diverse samples, the template-based method ensured consistency and alignment with the original dataset.
> - Historical Continuity: As you correctly pointed out, the template-based expansion aligned with the method utilized for the original NEG-136-SIMP dataset. By using this approach, we maintained a degree of historical continuity and comparability between the original and expanded datasets.
> - Comparative Insights: Although direct comparisons between the two methods were not presented in the paper, our internal investigations suggested that both methods had their merits. GPT-3 generated samples offered variety and unpredictability, whereas the template-based method provided controlled variability based on predefined structures.
> - Generalizability: Both methods produced similar outcomes, which strengthens our confidence in the results.

---

### Official Review · Reviewer_yjSX · 2023-08-05

**Soundness:** 3

**Excitement:**

3: Ambivalent: It has merits (e.g., it reports state-of-the-art results, the idea is nice), but there are key weaknesses (e.g., it describes incremental work), and it can significantly benefit from another round of revision. However, I won't object to accepting it if my co-reviewers champion it.

**Paper Topic And Main Contributions:**

Work described in this paper extends  two severely underpowered datasets which were used to support conclusions that may not have been statistically sound.  Results show rankings of language models evaluated on these extended datasets differ from rankings on the original datasets.

**Reasons To Accept:**

The paper corrects two severely underpowered datasets which were used to support conclusions that may not have been statistically sound.

**Reasons To Reject:**

Related to soundness: No error ranges are shown and no statistical tests are performed on findings about bakeoff winners.  This would not be difficult: For example, the evaluation could use a per-item permutation test on the first and second place models of the original and extended datasets.  I'm not sure how widely these results are used in model selection, but if these rankings are important (and they are mentioned in the introduction), they should be tested for significance.  I imagine differences of a few percentage points given only a few dozen datapoints in the original experiments would have failed significance, had it been tried, and alerted the original experimenters to the insufficient power of the data.

Related to excitement: The rankings shown in Table 1 don't change much.  If I am reading this right, in all cases, the first place model as ranked by the extended dataset places no worse than second in the original ranking.  Again, if these rankings are important, the best models were not grossly misrepresented in the original experiments.

**Reproducibility:**

4: Could mostly reproduce the results, but there may be some variation because of sample variance or minor variations in their interpretation of the protocol or method.

**Reviewer Confidence:**

3: Pretty sure, but there's a chance I missed something. Although I have a good feel for this area in general, I did not carefully check the paper's details, e.g., the math, experimental design, or novelty.

---

> ### Author Rebuttal · Authors · 2023-08-29
>
> **R1. “No error ranges are shown and no statistical tests are performed on findings about bakeoff winners”**
>
> We have performed power analysis using McNemar's test on the accuracies of the top2 models for each dataset. Our analyses revealed the following:
> - We saw very low power for ROLE-88 (0.255) and NEG-136 (0.014)
> - For ROLE-1500, the power to differentiate between the top two accuracy models significantly improved to 0.721
> - For GPT3-generated NEG-1500-SIMP, the power reaches a full 1.000 in distinguishing top models
> - However, for the template-generated NEG-1500-SIMP, the power remained low: 0.043 for the top two models and 0.230 for the top 1 vs. top 3.
>
> From these results, it is evident that ROLE-1500 and NEG-1500 (GEN) are dramatically more reliable datasets. Specifically, they can distinguish between small effects of approximately 0.03 for ROLE and about 0.15 for NEG.  Note that 0.15 is the smallest difference we could find in the real data, with a simulated power for a 0.03 difference in ROLE being 0.933.
>
> We also conducted a permutation test to check the statistical significance of the differences for extended datasets. The p-values for the top 2 models (referenced in Table 1) are as follows:
>
> - ROLE-1500: 0.0124
> - NEG-1500-SIMP-TEMP: 0.2206
> - NEG-1500-SIMP-GEN: 0.0002
>
> As you can see, there's a statistically significant difference for the models on ROLE-1500 and NEG-1500-GEN at the 0.05 significance level. In contrast, the template-generated dataset does not distinguish between the top 2 models. Thanks again for your feedback – we will include the above results in the final version of the paper.
>
> **R1. “The rankings shown in Table 1 don't change much”**
>
> The rankings indeed mostly (but not always) stay similar as we extend the dataset. But it's not just about rank, it's also about performance drop. For example, in the ROLE task, the top model doesn’t just move from first to second place. Its accuracy falls drastically from 55.7% to 26.1%. In other words, if we’re trying to answer a larger question of how well these models are able to do the ROLE task, the extended datasets show a significant change in model capability.
>
> With these additional details in mind, we'd appreciate if you could revisit your score.

---

### Meta-Review · Area_Chair_sFYp · 2023-09-18

**Recommendation:** 5

**Metareview:**

The work presents two valuable datasets on negation and role reversal and evaluate the performance of 22 models, showing that previous findings might have been skewed due to smaller test sets.

The three reviewers agree on their soundness and excitement scores, all on the upper bound, from good to strong/transformative, and the vast majority of comments and concerns are addressed on the rebuttal. These should be included in the final version of the manuscript.

---

### Decision · Program_Chairs · 2023-10-07

**Decision:**

Accept-Main

**Comment:**

The work presents two valuable datasets on negation and role reversal and evaluate the performance of 22 models, showing that previous findings might have been skewed due to smaller test sets.

The three reviewers agree on their soundness and excitement scores, all on the upper bound, from good to strong/transformative, and the vast majority of comments and concerns are addressed on the rebuttal. These should be included in the final version of the manuscript.